# Data-Driven, Short-Term Prediction of Charging Station Occupation

Roya Aghsaee [1,2,3], Christopher Hecht [1,2,4,*], Felix Schwinger [3], Jan Figgener [1,2,4], Matthias Jarke [3,5] and Dirk Uwe Sauer [1,2,4,6]

1 Grid Integration and Storage System Analysis, Institute for Power Electronics and Electrical Drives (ISEA), RWTH Aachen University, 52074 Aachen, Germany
2 Institute for Power Generation and Storage Systems (PGS), E.ON ERC, RWTH Aachen University, Matthieustr. 10, 52074 Aachen, Germany
3 Databases and Information Systems, RWTH Aachen University, 52074 Aachen, Germany
4 Juelich Aachen Research Alliance, JARA-Energy, 52056 Aachen, Germany
5 Fraunhofer Institute for Applied Information Technology FIT, 53757 Sankt Augustin, Germany
6 Helmholtz Institute Muenster (HI MS), IEK-12, Forschungszentrum Jülich, 52428 Juelich, Germany
* Correspondence: christopher.hecht@isea.rwth-aachen.de or batteries@isea.rwth-aachen.de;
Tel.: +49-241-80-493-66

**Abstract:** Enhancing electric vehicle infrastructure by forecasting the availability of charging stations can boost the attractiveness of electric vehicles. The transportation sector plays a crucial role in battling climate change. The majority of available prediction algorithms either achieve poor accuracy or predict the availability at certain points in time in the future. Both of these situations are not ideal and may potentially hinder the model's applicability to real-world situations. This paper provides a new model for estimating the charging duration of charging events in real time, which may be used to estimate the waiting time of users at fully occupied charging stations. First, the prediction is made using the random forest regressor (RF), and then the prediction is enhanced utilizing the findings of the RF model and real-time information of the currently occurring charging events. We compare the proposed method with the RF model, which is the approach's foundational model, and the best-performing prediction model of the light gradient boosting machine (LightGBM). Here, we make use of historical information of charging events gathered from 2079 charging stations across Germany's 4602 fast-charging connectors. To reduce data bias, we specifically simulate prediction requests for 30% of the charging events with various characteristics that were not trained with the model. Overall, the suggested method performs better than both the RF and the LightGBM. In addition, the model's structure is adaptable and can incorporate real-time information on charging events.

**Keywords:** electric vehicles; charging infrastructure; random forest (RF); ensemble learning

## 1. Introduction

Global warming has been a growing concern. In the Paris agreement, the participating countries have agreed to hold "the increase in the global average temperature to well below 2 degree Celsius above pre-industrial levels and to pursue efforts to limit the temperature increase to 1.5 degrees Celsius above pre-industrial levels" [1]. If the emission level exceeds the level that governments agreed upon in this goal, the irreversible consequences would threaten the global community. In order to adhere to this goal and based on the advice of the German Advisory Council on the Environment (SRU), the German government plans to reduce its greenhouse gas emissions by 95 percent less than the level of 1990, by 2050 [2]. To achieve this, the economy must be completely decarbonized; burning fossil fuels for energy must be minimized as much as feasible. All sectors are urged to reduce their greenhouse gas emissions drastically [2].

In Germany, the transportation sector generates the third largest share of greenhouse gas emissions after the energy and industry sectors [3]. Burning fossil fuels for cars is

the lion's share of transport-related emissions [2]. Since the German government aims to achieve carbon-neutrality in transportation by 2050, the transport sector requires immediate change [2].

A generally accepted solution of lowering $CO_2$ emissions and combating climate change is the use of electric vehicles (EVs). They are more eco-friendly substitutes for conventional internal combustion engine vehicles and produce less air pollution [4].

As the increasing usage of EVs becomes more apparent from one year to another, the imposed challenges need to be addressed. One of the unavoidable outcomes of the higher number of EVs is that there is a need for available charging infrastructure and expansion of charging infrastructures. This enables EV drivers to charge more comfortably and reduces the waiting time [5,6]. The governments should continue to support the building of new charging stations, specifically fast chargers, so that the on-road electrical vehicles can be served [7]. To ensure that nobody is left behind, fair access to charging infrastructure should be provided [7]. Fast charging was practically non-existent in 2013 [8], and has only become a strong growth-domain in the last few years with more and more vehicles able to draw such high powers [9]. While there has been an increase in the overall number of charging stations, because of the high investment cost of fast chargers, the number of them is still limited.

Another challenge is that the growing number of EVs requires the charging infrastructure to become more intelligent. The proportion of battery electric vehicles (BEVs) to charging stations is continuing to increase [10]. This means day by day, there are more BEVs that need to be charged at each station. This increasing trend requires smarter charging infrastructure. The growing number of BEVs needing to be charged and the user's lack of information on station availability may cause drivers to wait in a queue for hours at a station that was available minutes before [4].

To tackle the aforementioned problem, there are multiple approaches. A solution is to vary prices so that the users would avoid long parking times, or encourage them to use cheaper, less crowded stations [11,12]. Pricing techniques have been used in smart charging, to prevent grid load peaks, by using the current energy prices and considering varying usage between day and night and between weekdays [13,14].

Another approach to deal with this problem is to organize the limited existing charging infrastructures to be able to charge fleets of EVs with limited infrastructure capacity [15–18]. For example, workplace charging stations that are responsible for charging a fleet of vehicles, should intelligently prioritize charging specific EVs based on their properties, such as parking time and their current state of charge [15,19]. Similarly, the limited resources need to be assigned between different stations fairly [5,15,19].

Waiting time prediction at stations is another helpful, yet less taken approach. This helps EV drivers to know if each station is available at each point in time, and if not, predict the waiting time. Predicting charging station occupation status has several benefits. On one hand, users can benefit from the comfort of knowing the occupation status of different charging stations at their arrival time [4,5]. On the other hand, this enables charging station operators to manage stations and mitigate the congestion at stations and decrease the user's waiting time by balancing the load of EVs on different stations. This paves the way to create a real-time charging station recommendation system for EVs [4,20,21].

Additionally, it can facilitate the creation of applications that are directly integrated into the user interface of the car or on the user's smartphone to minimize the amount of time the vehicle is left inactive when charging sessions are terminated. These applications depend on precise short-term EV charging station waiting time forecasts for the next few minutes to hours [4].

Reviewing recent publications that have used data-driven approaches to predict the occupancy of EV charging stations, several approaches towards the aforementioned challenges can be identified.

Zhiyan Yi et al. [22] demonstrate a deep learning model-based approach for forecasting the demand for electric vehicle (EV) charging. The authors argue that for effective and

economical planning and management of EV charging infrastructure, precise demand forecasting is essential. The long short-term memory (LSTM) network is used in the proposed deep learning model to identify patterns and connections between past charging data and anticipated future charging demand. In order to increase the model's accuracy, the authors also employ a feature engineering technique to extract valuable features from the charging data. Overall, this study adds to the expanding body of knowledge on EV charging demand forecasts and shows how deep learning models have the potential to increase the effectiveness of EV charging infrastructure.

Zhu et al. [23] present a comparative study of different deep learning approaches for forecasting the electric vehicle (EV) charging load. The authors argue that accurate load forecasting is essential for efficient and reliable operation of EV charging infrastructure. The study compares several deep learning methods, including long short-term memory (LSTM), convolutional neural network (CNN), and gated recurrent unit (GRU), for load forecasting. The authors use real-world data from a public EV charging station in Singapore to evaluate the performance of these methods. The results show that the LSTM model outperforms the other models in terms of accuracy and robustness. However, the authors note that the performance of the deep learning models is sensitive to the length of the training data and the size of the input window.

In [4], the authors propose a hybrid LSTM neural network approach for multistep occupancy prediction at EV charging stations. Their approach combines both the univariate and multivariate LSTM models, along with a sliding window approach to capture the temporal dependencies and fluctuations of occupancy patterns. They demonstrate that the proposed hybrid LSTM model outperforms other baseline models in terms of prediction accuracy.

In [24], authors present a deep learning approach for long-term prediction of EV charging station availability. They use a combination of LSTM and MLP models to predict charging station availability over multiple time horizons. The authors show that their approach outperforms traditional statistical models and other machine learning methods.

In [25], the authors propose an ensemble machine learning algorithm to predict the charging duration time of EVs. They use the Shapley additive explanations method to interpret the results of the ensemble model and to identify the most influential features in the prediction. The proposed approach is shown to be effective in predicting charging duration times for a large data set of EV charging sessions.

In this study, we extend ideas presented in the literature, particularly with regard to ensemble learning models. Specifically, we aim to predict the short-term occupation status of charging stations utilizing the random forest (RF) [26,27] ensemble learning method for each station. As an alternative to the default RF regression procedure, which uses the average of all trees' predictions as the final prediction, a method is proposed in which we use each estimator's prediction combined with the real-time information of the current status of the charging point to calculate the most probable charging duration. Thus, at the moment a driver wishes to go to a station, the waiting time at the station until at least one charging point within the station would be available may be estimated. This waiting time estimate can also be produced for neighbouring stations, allowing drivers to select the station with the shortest waiting time. The approach offered for this prediction is tested on the real-world data of approximately 4600 fast-charging points in Germany.

With our study, we provide a proven methodology to enable accurate waiting time prediction that can be computed with limited computational resources and with high speed. This means that the developed algorithms can be employed in applications running on smart phones or other small computers, thereby working towards the goal of simplifying the user interface, as stated earlier. Due to the RF, the model is highly explainable, while retaining accuracy by incorporating real-time information without the need to retrain models for later time steps.

From these considerations, the following research question is distilled, which will guide the content of the remaining paper:

How can the machine learning tools be employed to efficiently predict how long a charge point at an electric vehicle charging station will be occupied?

## 2. Materials and Methods

This section introduces the materials and methods used in this paper. Given its data-driven approach, we first start by introducing the data used and subsequently focus on the methodology applied.

### 2.1. Data

This section starts by outlining the data used. Subsequently, we show how this data was transformed and divided in to a training and a test set.

#### 2.1.1. Historical Data

The industry partners SMART/LAB and Hubject have provided the charging station occupation to the ISEA as part of the BeNutz LaSA project (https://benutzlasa.de/, accessed on 15 December 2022).

The data under consideration refer to 4602 fast-charging points all over the Germany. The data set contains data from 1 January 2021 to 1 January 2022, for which the charging stations status changes are available. The aspects of the historical data that will be used in this work contains the following tables' structure in the database:

1.  Charging stations (referred to as Parks in the database) and their characteristics (latitude, longitude, address).
2.  Electric vehicle supply equipment (EVSEs) inside each charging station.
3.  Outlet type and power of EVSEs (EVSE_connectors).
4.  Status changes of EVSEs and the date and time of the availability and occupation (EVSE_status).

Figure 1 shows the class diagram of the historical data tables and their relations in the database.

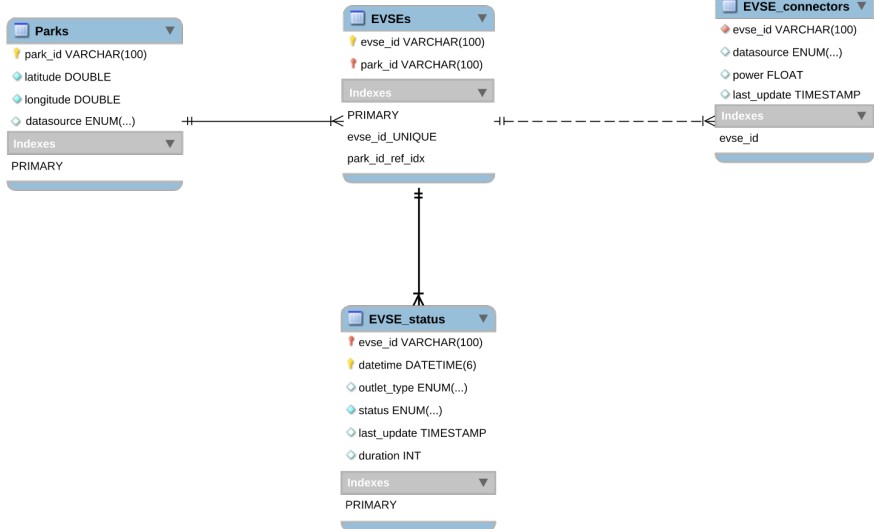

**Figure 1.** The structure of the historical data. Each charging station may contain multiple EVSEs. Similarly, for every entry in the EVSEs table, there can be one or more entries in the EVSE_connectors table, as the power of the connector can change after a while to a higher-level connector. The EVSE_status table keeps track of all status changes for all connectors in the EVSEs table. The status of each charging point (EVSE) is either available or occupied, and the duration column shows the duration that the charging connector has remained in the respective status in minutes.

### 2.1.2. Feature Selection

Finding the ideal features for the machine learning model and eliminating unimportant information that does not impact or improve the output of our model are the processes involved in feature selection. The following is a list of the features that are used to forecast how long charging events will last:

- Station location: We assume neighboring stations have a roughly comparable occupancy. It is also a way of differentiating between different stations without using individual IDs.
- Start time: EV users who start their charging sessions early in the morning may also be rushing to get to work. Hence, how long drivers stay at a station varies in the early morning or late evening.
- Month: The month of the charging event takes into account any seasonal influence.
- Power: Despite the fact that the data is limited to charging events of fast chargers, it is anticipated that recharging at a connection with 95 kW of power takes longer than one with 150 kW.
- Weekday and IsWeekend: The weekly average of station occupation is shown in Figure 2 for various times of the day. The occupation average for various days of the week is depicted in the right diagram. It is clear that the weekend's (Saturday and Sunday) occupation is both quite similar to each other and distinct from that of the other days of the week during rush hours (after 10 AM to 16 PM). Additionally, other days of the week have occupation distributions that are more comparable to one another than the weekend, although other days nevertheless have distinct occupation average distributions. The average occupancy rate for a day, divided by weekday, is shown in Figure 2a. Figure 2b compares the average occupation on weekends and other days of the week. The figures display normal charging behavior, including daily cycles and higher usage before and after working hours on weekdays but lower usage during working hours compared to weekends. Generally speaking, all stations follow these trends.
- Occupation_avg: The average occupation, which is the output of the average week model, is used as an input feature. The average week model is a basic model used by Hecht et al. [5] as a baseline for the prediction of mid-term occupation status. It is calculated by averaging the weekly usage of a charging station at an hourly resolution. For this work, the averaging is carried out for each week of the year, considering 0 for available status and 1 for occupied. Equations (1) and (2) show the average week model process, where $d$ is the day of the week, $h$ is the hour of the day and $w$ stands for the week number.

$$X_{d,h,w} \ with \ d \in \{Mo, \dots, Su\} \land h \in \{0, \dots, 23\} \land \omega \in \{1, \dots, 53\} \tag{1}$$

$$P_i(d, h, w) = \frac{\sum x_i(d, h, w)}{count(x_i(d, h, w))} \ for \ each \ i \tag{2}$$

Including occupation_avg, alone improves prediction performance significantly.

### 2.1.3. Train–Test Split

For the purpose of evaluating a machine learning model, data is often split into a training set and a test set. This was similarly accomplished in this study by using 30% of the data for testing and the remaining 70% for model training.

We use the *train_test_split*() function to divide the data. The shuffle parameter of the function divides the data randomly into training and test sets. Since the usage pattern of charging stations may change over time, it can be ensured that the models have not just been evaluated on very old or very recent events and that they are generalizable. For example, the model that has a very good evaluation performance on the data of the initial peak of the coronavirus cannot be trusted to have promising prediction results during

the current situation. The model should be retrained frequently in order to adapt to the changes in the charging infrastructure usage trends.

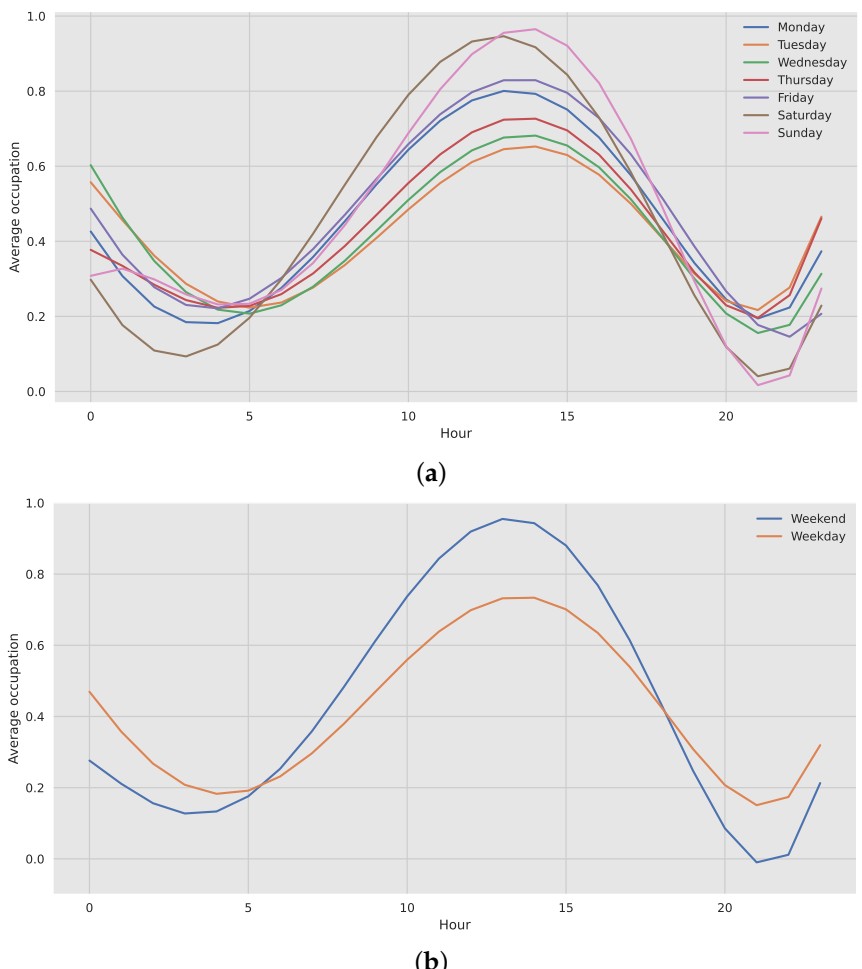

(a)

(b)

**Figure 2.** Average distribution of occupations during a day, separated by weekday (**a**) and weekend (**b**). It should be noted that the occupation average has been normalized between 0 and 1.

## 2.2. Methods

The procedure begins with selecting the suitable machine learning regressor model and continues with explaining how the selected model is used and enhanced so that it better corresponds to the real-time nature of the prediction. The steps involved in this improvement are broken down into specific details.

### 2.2.1. Model Selection

The prediction of charging duration at charging stations from the features is a regression task. The PyCaret library is used to compare the performance of several models to forecast the charging duration of charging events and to select the appropriate regression models for the available data set. Eighteen different regression models have been evaluated and based on the evaluation results, which are depicted in Table 1; two ensemble learning models of the light gradient boosting machine (LightGBM) and RF are proven to have the best performance. Note that PyCaret's compare_models() does not perform hyperparameter tuning. This means that models may be subject to overfitting and that performance gains are still possible. Nevertheless, the untuned models offer a fair ground for comparison.

**Table 1.** The output of PyCaret's model comparison across the entire data set using 10-fold cross-validation. The highlighted values correspond to the best score per metric.

| Model | MAE | MSE | RMSE | $R^2$ | RMSLE | MAPE | TT (s) |
|---|---|---|---|---|---|---|---|
| lightgbm | 21.0785 | 1886.5135 | 43.4328 | 0.1139 | 1.1626 | 3.9587 | 0.9570 |
| rf | 21.6954 | 1968.1448 | 44.3622 | 0.0756 | 1.0931 | 3.3667 | 58.5450 |
| gbr | 21.4235 | 1969.9957 | 44.3835 | 0.0747 | 1.1816 | 4.0781 | 22.8000 |
| lr | 21.9907 | 2067.0375 | 45.4635 | 0.0292 | 1.2245 | 4.4208 | 1.0910 |
| ridge | 21.9907 | 2067.0375 | 45.4635 | 0.0292 | 1.2245 | 4.4208 | 0.0630 |
| lar | 21.9907 | 2067.0375 | 45.4635 | 0.0292 | 1.2245 | 4.4208 | 0.0600 |
| br | 21.9906 | 2067.0375 | 45.4635 | 0.0292 | 1.2245 | 4.4208 | 0.1760 |
| omp | 21.9302 | 2089.3212 | 45.7078 | 0.0187 | 1.2245 | 4.3805 | 0.0600 |
| en | 22.3014 | 2110.1957 | 45.9356 | 0.0089 | 1.2419 | 4.5140 | 0.1830 |
| lasso | 22.3314 | 2114.2666 | 45.9799 | 0.0070 | 1.2427 | 4.5161 | 0.2090 |
| llar | 22.3500 | 2129.1706 | 46.1417 | −0.0000 | 1.2428 | 4.4775 | 0.0700 |
| dummy | 22.3500 | 2129.1705 | 46.1417 | −0.0000 | 1.2428 | 4.4775 | 0.0390 |
| huber | 20.8329 | 2144.1070 | 46.3031 | −0.0070 | 1.1417 | 3.3577 | 2.2830 |
| et | 22.7264 | 2167.9740 | 46.5599 | −0.0183 | 1.1494 | 3.2787 | 38.8670 |
| par | 24.1666 | 2243.1577 | 47.3323 | −0.0537 | 1.2837 | 3.9364 | 0.2330 |
| knn | 23.9175 | 2270.0498 | 47.6439 | −0.0662 | 1.1837 | 3.7279 | 159.2090 |
| ada | 37.0643 | 3298.0843 | 57.4235 | −0.5492 | 1.5346 | 8.3365 | 4.8830 |
| dt | 28.4673 | 3802.5382 | 61.6637 | −0.7862 | 1.3677 | 3.3913 | 1.8300 |

The columns of Table 1 represent common metrics used in regression issues. These are mean absolute error (*MAE*), mean squared error (*MSE*), root mean squared error (*RMSE*), R-squared ($R^2$), root mean squared logarithmic error (*RMSLE*), and mean absolute percentage error (*MAPE*). Since these are common machine learning metrics, a thorough discussion of each is excluded at this time and the reader is directed to the paper's Appendix A.

Of the analysed models, LightGBM and RF are consequently chosen as models for further analysis. Since PyCaret does not perform a parameter optimization, this was subsequently carried out. For RF, parameters were chosen manually to ensure accuracy while avoiding overfitting as follows: criterion = 'mse', max_features = 'auto', min_samples_leaf = 8, min_samples_split = 5. All other parameters were left in the default settings of sklearn. For the LightGBM, a grid-search for hyperparameter tuning was performed for the key parameters num_leaves, max_depth, and learning_rate. The resulting values were learning_rate = 0.1, max_depth = 60, and num_leaves = 120. All other parameters were again left as default.

### 2.2.2. Prediction

The approach is to provide a technique for improving the model by making use of the real-time nature of the task and including some new information into the model. To improve the prediction, it is intended to leverage the individual inducers of ensemble models as well as the readily available real-time information on how long the EVSE in the target charging station has already been occupied. The proposed prediction enhancement method is suitable for the parallel ensemble model category, hence the RF model is used. After detailed explanation of the method, it will be made clear why the sequential category of ensemble models is not appropriate for the improvement that is being provided. However, as the LightGBM model performed better than the base RF model, we also made the prediction using LightGBM to show that the suggested fine-tuned RF method works better than the promising LightGBM.

An iterative, multi-step technique is used for the prediction. The prediction algorithm is summarized in the activity diagram in Figure 3. The algorithm always bases its prediction on a particular charging station. Additionally, the algorithm forecasts charging durations for each EVSE separately. A charging station is typically occupied if all of the EVSEs inside of it are being used.

The model forecasts how long various EVSEs sessions will take to charge inside the desired charging station. The remaining duration of the occupancy of each EVSE at the time of prediction is important since it determines how long a user must wait to begin

charging at the station. In conclusion, the prediction system determines how long each station's EVSEs will take to charge based on real-time data from those units. The algorithm then incorporates each individual EVSE's prediction into the final prediction.

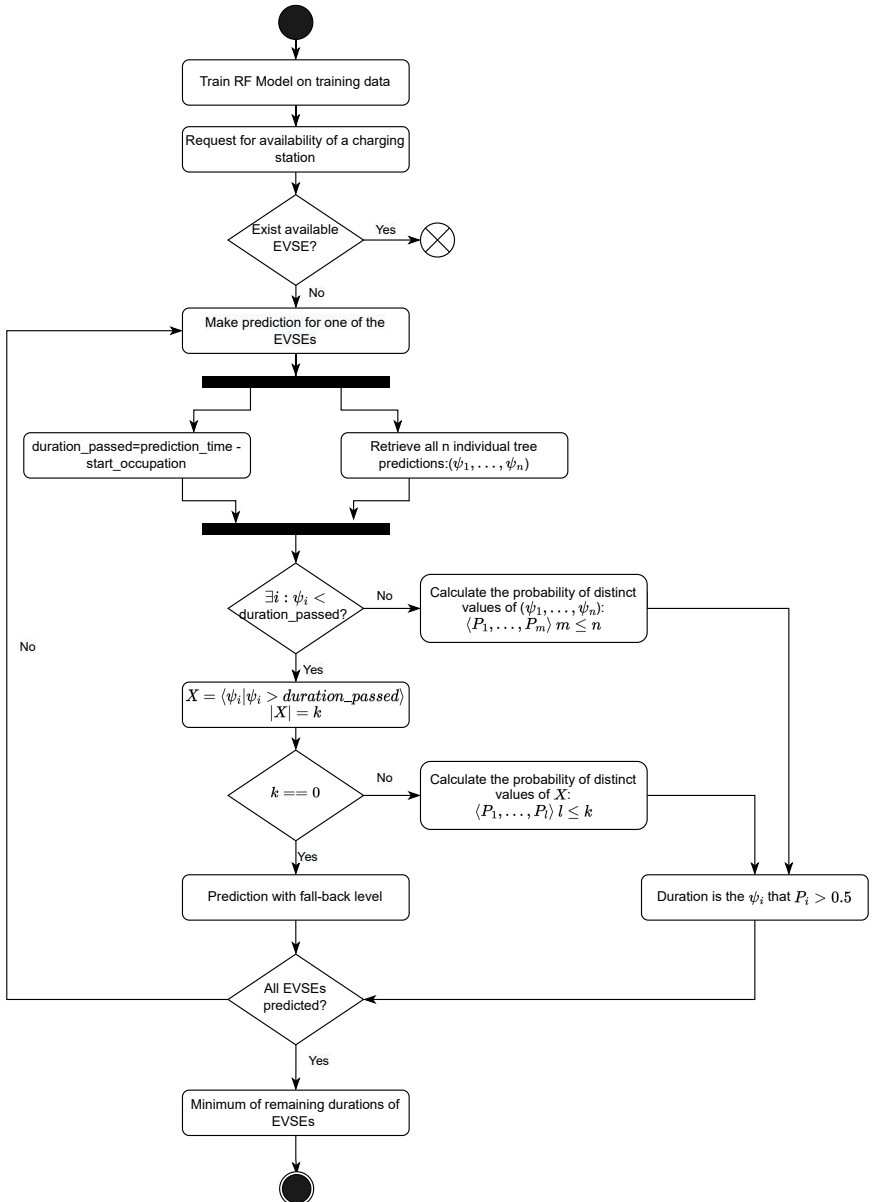

**Figure 3.** The forecast flow in general. The RF model is initially trained. The prediction is then made for each EVSE separately when a driver wishes to travel to a charging station that is completely full. All of the EVSEs in the station go through the procedure iteratively, and the results are then combined to estimate how long the user will have to wait.

The process depicted in Figure 3 is broken down into its component parts and elaborated upon in the following example.

- In the first step, the RF model is trained using the complete historical data of charging events as well as relevant features, along with 100 estimator trees. The hyperparameter tuning method is used to determine the number of estimators that the RF model will have.
- Assume that a driver plans to arrive at a fully occupied charging station with two EVSEs at time $t_1$, so that they can recharge their battery. As a result, the driver wants to find out whether or not the station has available EVSEs and, if it does not, how

much they will have to wait before they can begin charging. Taking into account the fact that both of the EVSEs are being utilized at the given time $t_1$, the following method of duration prediction will be carried out on each of the EVSEs separately, until the charging duration of the running events at each of the EVSEs in the target station is predicted.

- Consider the time $t_0$ to be the beginning of the charging event that took place at the EVSE. The duration that the station has been occupied up to this point can be determined (*duration_passed* $= t_1 - t_0$).
- The trained RF model will be used to make a prediction for the total charging time that will be required for the charging event. Each of the estimators that make up the RF model makes a prediction regarding the duration, and the RF model provides the average of the values that are anticipated. However, we take the value that was predicted for each tree. $\psi_i$ represents the outcome of the *i*th tree analysis.
- We know how long the EVSE has been occupied and have the results of estimate tree predictions. Consequently, it is evident that the trees that predicted a duration shorter than *duration_passed* are incorrect. To improve the model's prediction accuracy, we filtered out the incorrect predictions.
- The charging period of the present event at the EVSE is the median of the predictions made by the remaining, more accurate trees. The median is far less affected by extreme values than the arithmetic mean. As consumers occasionally remain at a charging station for far longer than usual, the estimate trees always include some very lengthy durations that seldom occur. Therefore, the median performs better than the mean. On the other hand, if the events at a particular station are likely to be very lengthy, the majority of estimators predict that the duration will be long, and the median is likewise an accurate representation.

  To describe it formally, assume $\langle \psi_1, \psi_2, \ldots, \psi_n \rangle$ represents the ordered list of various duration estimates given by $n$ estimator trees of RF for a certain charging event, with $\psi_1$ being the lowest forecast and $\psi_n$ being the greatest. The unique prediction values are $\langle \psi_1, \psi_2, \ldots, \psi_m \rangle$ where $m \leq n$.

  The probability list $\langle P_1, P_2, \ldots, P_m \rangle$ represents the likelihood of a unique prediction exceeding the anticipated values. For instance, $P_1$ is the probability that the duration will be greater than $\psi_1$ and is always 100%. $P_k$ indicates the likelihood that the duration will be at least the expected amount where the median corresponds to $\psi_k$.
- For some charging occurrences, all estimators may have estimated the length to be shorter than the *duration_passed*. After filtering out the incorrect estimators, there would be no remaining estimator results to make the prediction upon.

  To predict the charging duration in such instances, models are trained for specific power levels, and if the station-specific model cannot make a reasonable prediction, we move to these broader models. The graph shown in Figure 4 depicts the length of time people stayed based on the starting time indicated on the x-axis. The blue shades indicate the deciles. For events that are longer than usual, the median of the remaining portion is utilized to predict the duration. Therefore, if 10 h have passed and that corresponds to 20% of the data, then the new expected value is the median of the remaining parts, which is the 40% decile $((100\% - 20\%)/2)$.

- The remaining wait time at the station is the minimum of the durations determined for the various EVSEs within the station. The new user is able to begin charging at the station as soon as one of the EVSEs becomes available.

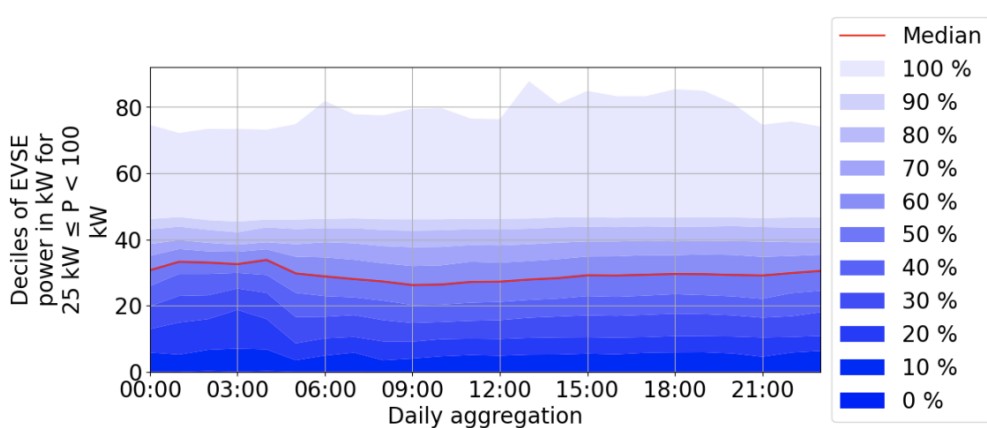

**Figure 4.** The fall-back model of the power level of 25–100 kW. Taken from [10].

RF and LightGBM, two ensemble learning methods, produce promising outcomes in terms of regression metrics. The sequential ensemble methods cannot be used for the proposed method of this study, as in the sequential method of gradient boosting, trees are generated sequentially and each tree is constructed based on the errors of the preceding tree to enhance the outcomes at each step. Therefore, it is not possible to filter out the estimators that are intended to improve the outcomes of the succeeding estimator.

Consequently, the key difference between our suggested approach and conventional RF and LightGBM is that we use the real-time information that becomes available after the start of the charging event. By removing trees for which we already know the prediction is wrong, the prediction accuracy can be dramatically improved for long charge events, while only using information that is actually available at the time of prediction.

## 3. Results

This section provides a thorough analysis of the results of the prediction approach and discusses the evaluation results of two different evaluation scenarios. Additionally, in order to make comparisons, this section compares the prediction model of this paper with two other prediction models, RF and LightGBM. In order to demonstrate that there has been an improvement in performance, it is useful to compare the final results with the RF regressor model, which is the foundation for the prediction approach of this work. In addition, out of the twenty-four different typical regression models, LightGBM emerged as the one with the most promising metrics. In order to provide evidence that the proposed model is superior to the most accurate predictive model when applied to real-world situations, the results of the prediction are compared to this model.

### 3.1. Model Results

This section investigates the outcomes of the real-time RF model based on two different request simulation methods. In both methods, the request intervals of 1, 2, 5, 10, 20, and 30 min are considered, but with two different interpretations.

#### 3.1.1. First Evaluation Scenario

In the first scenario, the user requested to check the availability of a charging point $m$ minutes prior to the charging point becoming available (as indicated by the $m$ minute request interval), in other words, $m$ minutes prior to the charging point's status change, to determine how accurate the forecasts are just before the change in occupation status.

Figure 5 depicts the first scenario's results confusion matrices for different request intervals. The expected and actual values are categorized. The rows represent the actual categories, while the columns represent the predicted ones. The outcome is normalized to the actual values (rows). The values on the main diagonal indicate the events whose

duration category was accurately anticipated. The first statistic of the matrix corresponding to the request interval of 1 min indicates that 0.49 of the events whose actual occupation duration was less than 10 min were anticipated to have a duration of less than 10 min. A request interval of $m$ minutes means that the charging point is occupied $m$ minutes before it becomes available. Therefore, its actual duration is more than $m$ minutes. This explains the 0 values above the main diagonal. For example, for a matrix corresponding to a request interval of 30 min, there are no charging events with actual charging duration less than 30 min.

Figure 5 also depicts that the closer the request is to the end of the charging event, the more the majority of events concentrate along the main diagonal and fall into their actual category or one closely related to it. Contrarily, the forecast accuracy decreases with increased request intervals. This can also be due to the fact that these longer intervals correspond to longer charging events, and as the majority of events are relatively short at fast-charging points, the prediction accuracy for these events drops. Among others, we can see that the majority of the values below the main diagonal are zero. This demonstrates that the predicted duration of infrequent charging occurrences is shorter than their actual duration. This behaviour can be considered desirable since long events are seldom.

The lack of accuracy for $m = 30$ can also be explained by the fact that as most charging events are short, the models are generally optimised towards short event durations.

Figure 6 shows the error distribution of different request intervals before the end of charging occupation for the events in the test data. According to the error distributions, the 0 error occurs frequently and is generally negative for lower request intervals, indicating that expected durations are typically higher than actual durations. The reason for this is that the approach eliminates the predictions that are shorter than the passed durations. On the other hand, for longer request intervals, the remaining duration is higher and the passed duration is shorter. As a result, fewer incorrect predictions are eliminated, increasing the number of positive errors (predictions less than the real values). These diagrams accurately show that the longer the passed duration, the higher is the prediction accuracy.

### 3.1.2. Second Evaluation Scenario

In the second scenario, using the $m$ minute request interval, the user asks to check whether a charging point is available $m$ minutes before she arrives at the station. As the users typically seek the availability of a charging station shortly before they arrive, it is of special importance to assess the performance of the model for short request intervals.

Table 2 shows the comparison of the prediction power of the real-time RF model for the different request intervals before arrival of the user at the station. Surprisingly, for longer request intervals, the prediction performance outperforms the shorter request times. The longer the user checks before arriving at an occupied fast-charging station, the more probable it is that the station will be available, considering the short charging durations at fast stations. On the other hand, a different user might show up at the station during this time and begin charging, resulting in the anticipated vacant station becoming filled. To confirm this idea, the Table 3 shows the values of the selected evaluation metrics after removing the charging events for which both the real and predicted statuses are available. It shows that the prediction performance of the model shortly before arrival of the user is better than with longer request intervals.

**Table 2.** Comparison of the predictive power of the real-time RF model at various request intervals prior to the user's arrival at the station in minutes.

| Request Time | MAE | RMSE | MAD |
|---|---|---|---|
| 1 | 15.70 | 35.22 | 8 |
| 2 | 15.50 | 35.13 | 8 |
| 5 | 14.76 | 34.80 | 8 |
| 10 | 13.31 | 34.18 | 5 |
| 20 | 11.74 | 33.53 | 0 |
| 30 | 7.54 | 31.56 | 0 |

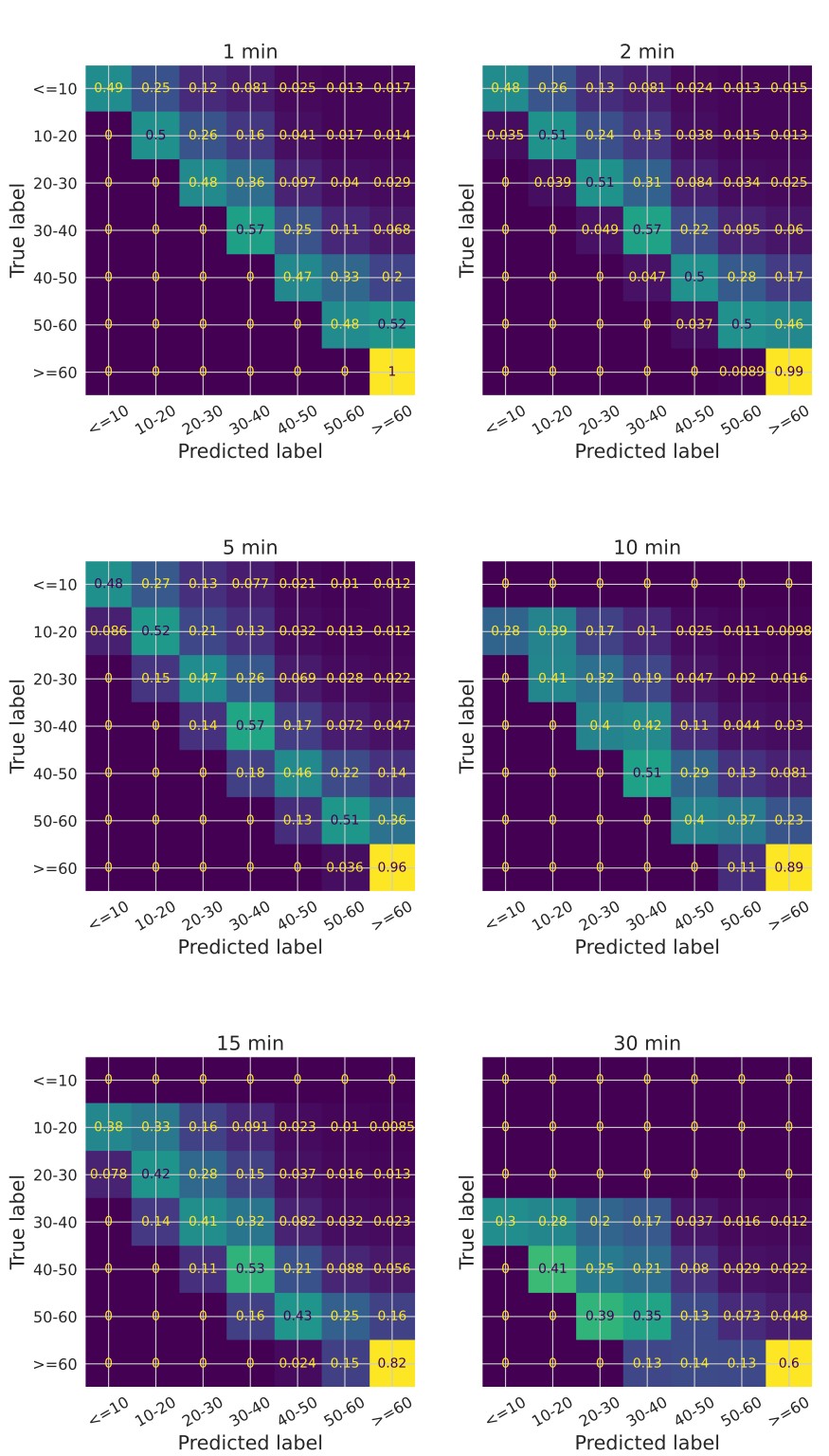

**Figure 5.** Confusion matrices of the prediction results for different request intervals. The labels of the rows correspond to the real durations and the columns indicate the predicted durations. The results are normalized over the true values (rows). Note that, although the models are regression models, we present confusion matrices to show the effect of the real-time components as well as to show for which types of predicted durations the model is particularly accurate.

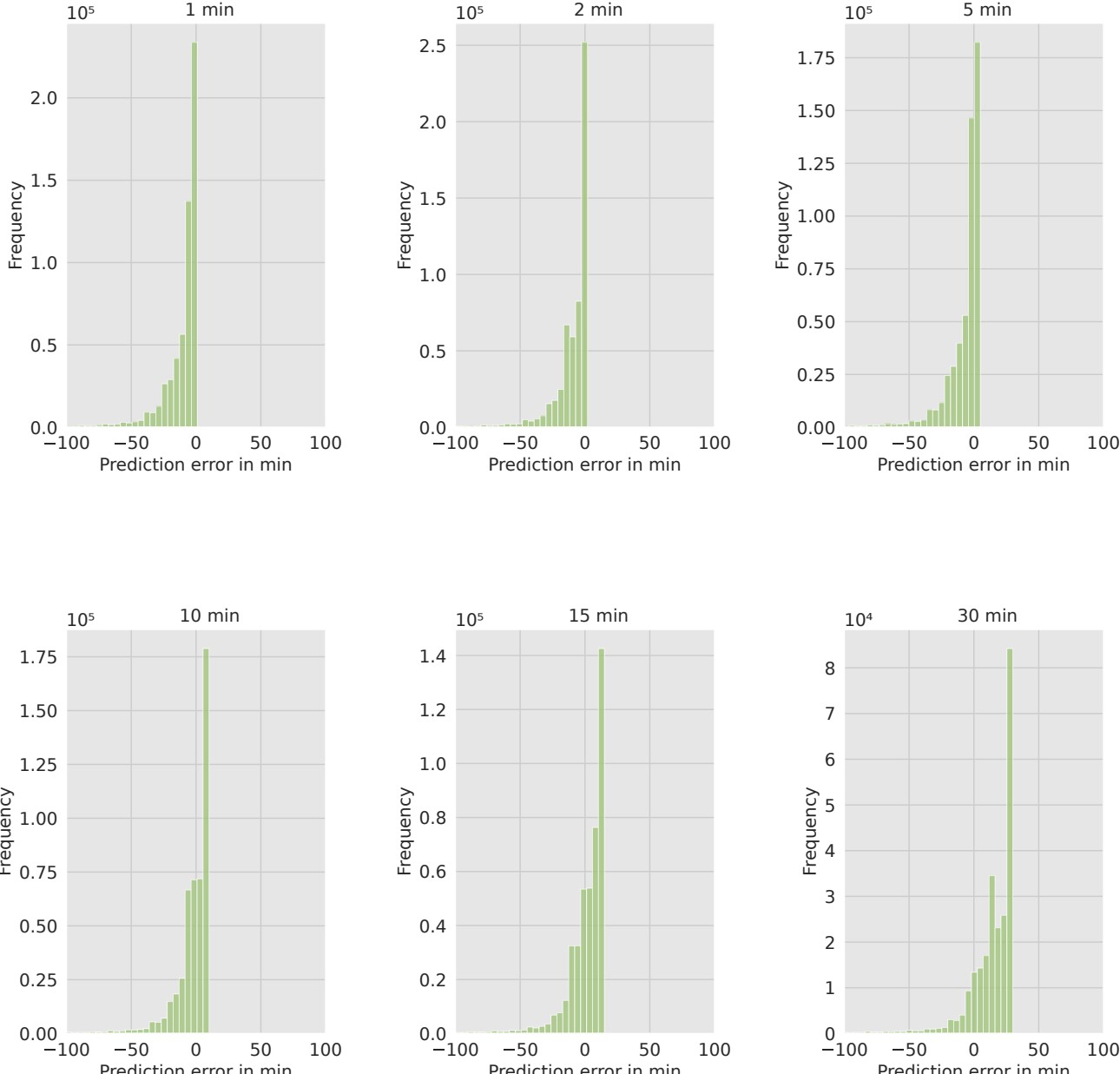

**Figure 6.** The error distribution of various request window intervals prior to charging occupation's termination. The error is computed by the real duration minus predicted duration.

**Table 3.** Comparison of the predictive power of the real-time RF model at various request intervals prior to the user's arrival at the station after removing the events that are available according to both prediction and real status in minutes.

| Request Time | MAE | RMSE | MAD |
|---|---|---|---|
| 1 | 16.98 | 36.62 | 10 |
| 2 | 17.48 | 37.3 | 10 |
| 5 | 18.48 | 38.94 | 11 |
| 10 | 20.28 | 42.13 | 12 |
| 20 | 22.79 | 46.72 | 14 |
| 30 | 36.28 | 69.69 | 20 |

## 3.2. Model Comparison

This section compares the approach model's results to those of the RF regressor and LightGBM models. This comparison is desired given that the RF regressor is the base for

the proposed model and LighGBM is the best performing model based on the findings of comparing the performance of eighteen alternative regression models.

Figure 7 displays the residual plots of the three prediction models of LightGBM, RF, and the model proposed in this work (real-time RF), respectively, from top to bottom, while employing both training and testing data sets.

The difference between the observed value of the target variable (actual charging duration) and the predicted value (predicted charging duration) are known as the residuals. The predicted values are shown on the x-axis of the residual plots, while the residuals are shown on the y-axis. The probability density function illustrating the distribution of the residuals is shown in the histogram on the right side of the figures.

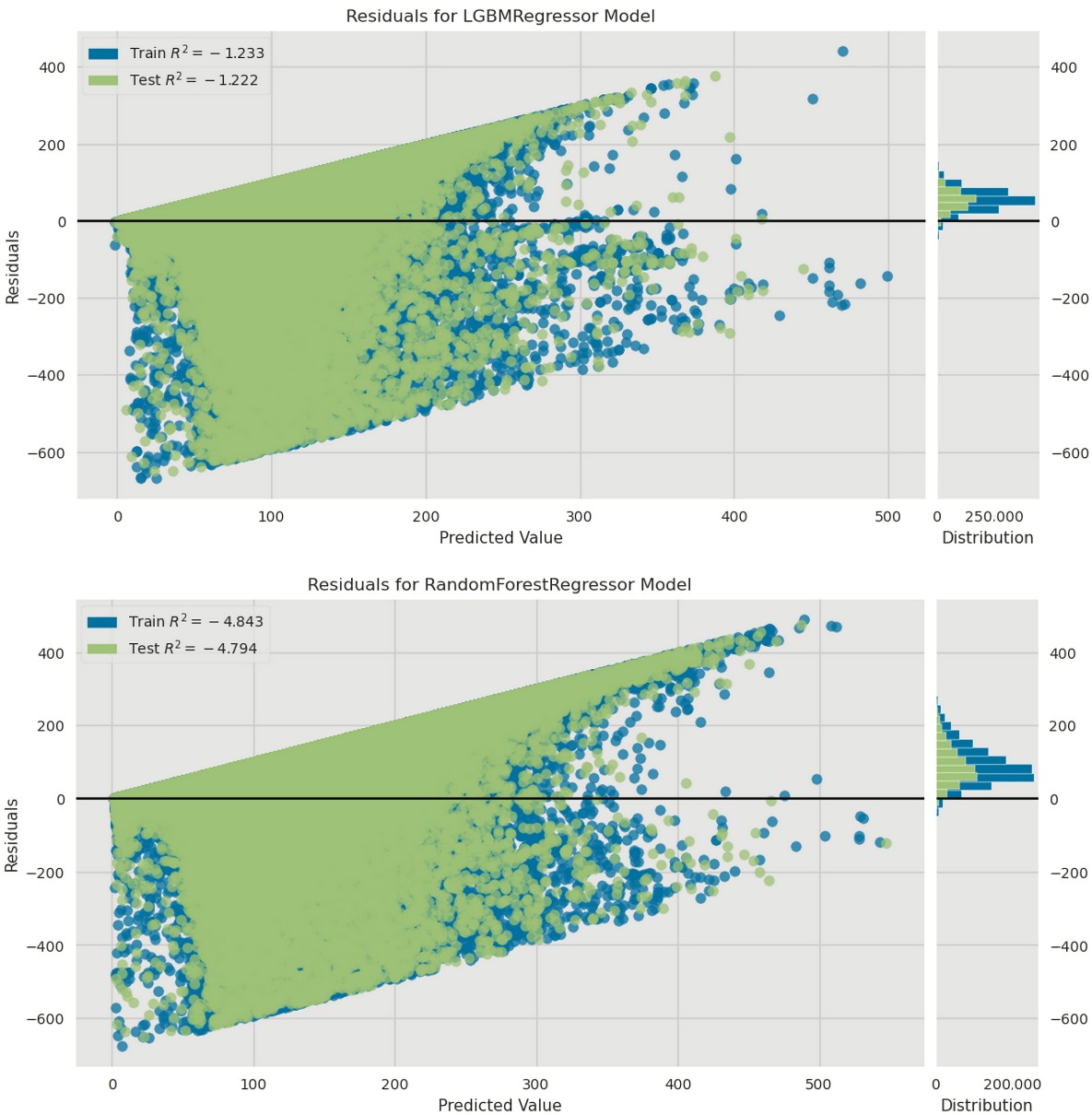

**Figure 7.** *Cont.*

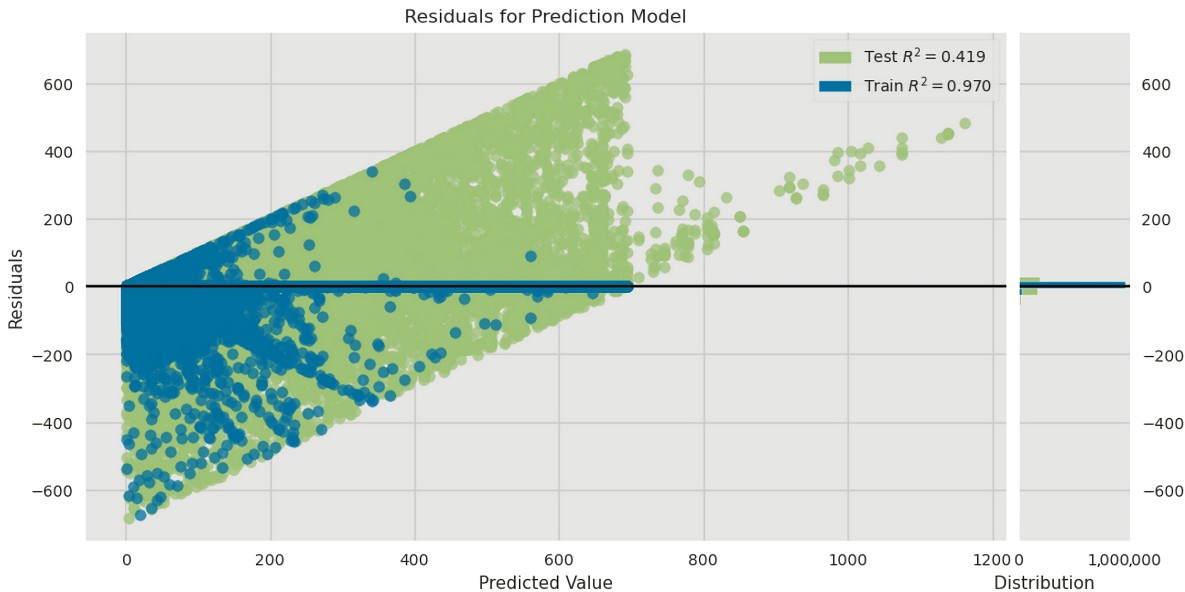

**Figure 7.** The residual plots of the three prediction models—LightGBM, RF, and real-time RF (top to bottom, first two on previous page)—using both training and testing data sets are shown. The residuals' distribution is depicted in the plots on the right side of each residual plot. Minutes are used for the residuals and predictions.

Each point in the plots represents a charging event, with the model's forecast on the x-axis and the model's prediction residual shown on the y-axis. How inaccurate the prediction was for that event is measured by the distance from the line at 0. Positive numbers for the residual (on the y-axis) for prediction (on the x-axis) indicate a prediction that was too short, while negative values indicate a prediction that was too long; zero indicates a prediction that was exactly correct.

These residual plots can be used to examine the regressors' error. The top and middle plot in Figure 7 both display a uniform distribution of errors in target variable prediction above the line at 0, indicating that charging event durations were typically predicted to be shorter than actual. The low distribution of residuals near 0 indicates that few charging events were accurately predicted. The reason for this phenomenon probably lies in the way that such models are typically trained: Long charge events are unusual, but if they happen, generate a large error. To improve the RMSE, the algorithm decides to put more weight onto these long events. This, however, leads to the average prediction departing from the median actual value, thus explaining the observed behaviour.

The bottom plot in Figure 7, on the other hand, depicts a relatively random and uniform distribution of residuals versus the target. It indicates that the predictive model is working effectively. The density plot on the right side also reveals that our model's error is normally distributed around zero, which shows the model is well fitted. This result is possible since the real-time approach allows us to handle long charge events differently and the algorithm consequently does not need to shift away from the median realized value as much. This demonstrates that the performance of the model's forecast has improved after deleting the incorrect estimators of the RF that anticipated the duration to be too short (in this example, shorter than half of the real duration). This logically caused the distribution of residuals, which in RF were generally above the line at 0 (shorter than actual duration), to be centred on 0.

Negatively, the real-time RF model's highest residual (bottom plot in Figure 7) is bigger than the RF model's prediction (middle plot in Figure 7). These high residuals cannot go above fifty percent of the real duration. As a consequence of this, it is clear that these significant errors correspond to charging events with a particularly lengthy charging duration. On the density plot, it is evident that the density of these points is not high.

## 4. Discussion

In this paper, we set out to explore how machine learning models can be used to predict how long a charge point will remain occupied. By comparing a total of eighteen models and creating our own custom model based on these findings, we are able to show the concept for an accurate machine learning model structure for the task at hand.

In terms of accuracy and overall performance of predictions, the recommended prediction approach is superior to other regression models, as previously demonstrated. In total, we analysed eighteen different regression models that are often used. In particular, it outperforms the two models of ensemble learning that are the best-forming models, LightGBM and RF. Because the forecast was constructed based on the results of the RF regressor model, it is still subject to the constraints that are associated with RF. In the following, this section discusses a number of limitations related to the evaluation and the model.

It was necessary to replicate real-time requests from consumers to approach already occupied charging stations in order to examine the suggested model. We implemented the simulation requests for the evaluation to take place. We evaluated the model based on two different prediction scenarios with two different assumptions. In the first scenario, we proved that there is a positive correlation between the amount of time that has elapsed from the charging events and the precision of predictions. This, however, comes with a shortcoming. This means if the user requests for the duration prediction of an occupied charging event at the beginning of the event, the prediction is not precise. In the second evaluation scenario, we predicted the status of the charging points at $m$ minutes before arrival of the user at the station. In this scenario, the evaluation metrics show that the longer before the arrival users ask for the prediction, the more accurate is the prediction. This shows another disadvantage of the model. When the users ask for the status prediction of a charging point long before their arrival, the model predicts the end of the current charging event and predicts that the charging point will be available afterwards. However, in real-world scenarios, the end of the current charging event does not mean that the charging point will be available at the time that the user arrives, since the charging point can be occupied by another user. In a real-world application, the proposed model would consequently have to be coupled with a model that predicts the likelihood of somebody else arriving in the meantime. Such a structure is proposed in [28] using the model proposed in this paper as well as the one from our previous publication [5] as a basis.

Another shortcoming is that the extremely long charging events are anticipated to be shorter than their real duration, despite the fact that the model performs well for short and moderately long charging events. This is because the algorithm has difficulty projecting extremely long charging events. These occurrences are likely to be outliers in the data; yet, customers require accurate information regarding the availability of charging stations and thus they expect the model to make accurate predictions in real-world scenarios. This could possibly have a significant effect on the consumers' choice of charging station and the amount of time they wait. Therefore, accurate prediction of lengthy charging events in applications that take place in the real world is of critical relevance. The issue is somewhat less problematic at charging stations with many EVSEs, since the likelihood of many extraordinarily long charge events at a singular station is extremely low.

We shuffled the data in order to divide it into a training set and a test set to avoid overfitting. A key reason for shuffling was that the coronavirus pandemic and the uptake of e-mobility at the same time resulted in a change of usage patterns over the observed time. By setting the overall time interval such that charging events occur at various times, this lessens bias. The training data now contains certain events that happened after some events in the test data as a result of this shuffling, which is a drawback, because the model in this case has knowledge about the future.

In real-time use cases, reducing the amount of delay experienced when serving predictions is absolutely essential, as the predicted action must take place immediately. The client only needs to send in one charging station in order to accomplish online forecasting. This user anticipates that the prediction will become available in less than a few millisec-

onds. In general, there are two different kinds of latency: one that occurs at the model level, and another that occurs at the serving level (https://cloud.google.com/architecture/minimizing-predictive-serving-latency-in-machine-learning, accessed on 17 December 2022). At the model level, latency is determined by the amount of time needed by the model in order to provide predictions. At the serving level, it depends on the amount of time that the system needs in order to provide the forecast in response to a request. Although the model's predictions are produced at a reasonable speed, in order to live up to the users' expectations for applications that take place in the real time, the model's processing speed should be increased.

## 5. Conclusions

In conclusion, this paper aimed to construct a short-term prediction model for charging events that incorporates real-time information for improved accuracy. The results of the study showed that the proposed model can provide accurate predictions; however, there are several limitations that need to be addressed before the model can be implemented in real-world scenarios.

Firstly, the model assumes that the available charging stations will be available until the user who intends to drive to the station arrives. This assumption does not always hold true in real-world scenarios, as a fully available charging station with two charging points can be occupied within a few minutes. Therefore, the model needs to take into account the real-time availability of charging stations.

Secondly, although the model's predictions are generated at a good speed, the model's processing speed should be enhanced to meet user expectations for applications that occur in real time. This is especially important for applications where users expect real-time predictions and might not be willing to wait for the model's predictions.

Thirdly, the model should be trained periodically so that it can adapt to changing trends and patterns. The data used in this study correspond to the charging events from 1 January 2021 to 1 January 2022. As the charging patterns and infrastructure are constantly evolving, the model needs to be trained with updated data to ensure its predictions remain accurate.

Finally, it is important to note that the data used in this study are specific to Germany and the results of the study may not generalize to other countries or regions. Therefore, further research is needed to evaluate the model's performance in other locations.

In summary, this study proposed a short-term prediction model for charging events that incorporates real-time information to improve accuracy. The model has the potential to be applied in both densely populated and remote areas, but further development is needed to make it suitable for real-world scenarios. The limitations of the model include the assumption of charging station availability and the need for enhanced processing speed, as well as the need for periodic retraining with updated data.

**Author Contributions:** Conceptualization, R.A., C.H. and F.S.; Methodology, R.A. and C.H.; Software, R.A.; Validation, R.A., C.H. and F.S.; Formal analysis, R.A.; Investigation, R.A.; Resources, C.H., J.F., M.J. and D.U.S.; Data curation, R.A. and C.H.; Writing—original draft, R.A.; Writing—review and editing, C.H., F.S., J.F. and M.J.; Visualization, R.A.; Supervision, C.H., F.S., M.J. and D.U.S.; Project administration, C.H., J.F. and D.U.S.; Funding acquisition, C.H., J.F. and D.U.S. All authors have read and agreed to the published version of the manuscript.

**Funding:** This research was funded by the Federal Ministry for Economic Affairs and Climate Action (BMWK) on the basis of a decision by the German Bundestag, grant number 01MV20001A.

**Institutional Review Board Statement:** Not applicable

**Informed Consent Statement:** Not applicable

**Data Availability Statement:** The data used in this project is protected by copyright and cannot be shared. A reader willing to reproduce our results is advised to contact a data broker that has the permission to share the occupation data of public charging stations in Germany.

**Acknowledgments:** We would like to extend our gratitude to the industry partners Hubject and SMART/LAB both for data provision as well as for guidance during the development of this work.

**Conflicts of Interest:** The authors declare no conflict of interest.

## Abbreviations

The following abbreviations are used in this manuscript:

| | |
|---|---|
| RF | Random forest |
| LightGBM | Light gradient boosting machine |
| EV | Electric vehicle |
| EVSE | Electric vehicle supply equipment |

## Appendix A

*Metric Descriptions*

In this section the metrics for regression tasks are discussed.

The *MAE* is an indicator of the average absolute divergence between the predicted values and the true values (charging duration). The error's unit is indicated in minutes. Real charging duration of the event $i$ is shown with $y_i$ and the predicted duration of the same event is $\hat{y}_i$, and $n$ is the number of observations.

$$MAE = \frac{1}{n} \sum_{i=1}^{n} |y_i - \hat{y}_i| \tag{A1}$$

The *MSE* identifies the average squared difference between the real duration and predicted duration. The *MSE* is equal to 0 when a model is error-free. Its value increases with the increase in the model error.

$$MSE = \frac{1}{n} \sum_{i=1}^{n} (y_i - \hat{y}_i)^2 \tag{A2}$$

The *RMSE* is defined as the residuals' standard deviation. Large residuals $(y - \hat{y})$ are penalized by the *RMSE* by using the square of the error value. Therefore, the greater the deviation between a charging duration prediction and the actual charging length, the greater the penalty. Errors are indicated in minutes.

$$RMSE = \sqrt{\frac{1}{n} \sum_{i=1}^{n} (y_i - \hat{y}_i)^2} \tag{A3}$$

The percentage of the variance in the dependent variable that can be predicted from the independent variable(s) in a regression model is depicted by the *R*-squared ($R^2$) metric, which is a statistical measure. In other words, it calculates the percentage of the observed data's variation that the model can account for.

$$R^2 = 1 - \frac{\sum_{i=1}^{n} (y_i - \hat{y}_i)^2}{\sum_{i=1}^{n} (y_i - \bar{y})^2} \tag{A4}$$

Root mean squared logarithmic error (*RMSLE*) is a metric used to evaluate the performance of regression models, particularly when you have a lot of varying magnitudes in the target variable. It is similar to the root mean squared error (*RMSE*), but the logarithm of the predictions and true values are used instead of the actual values. The formula for *RMSLE* is

$$RMSLE = \sqrt{\frac{1}{n} \sum_{i=1}^{n} (log(y_i + 1) - log(\hat{y}_i + 1))^2} \tag{A5}$$

Mean absolute percentage error (*MAPE*) is a commonly used metric for evaluating the performance of regression models. It measures the average percentage difference between the predicted values and the true values. The formula for *MAPE* is

$$MAPE = \frac{100}{n} \sum_{i=1}^{n} \frac{|y_i - \hat{y}_i|}{y_i} \tag{A6}$$

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
