# Peer review of "Data-Driven, Short-Term Prediction of Charging Station Occupation"

_electricity, doi:10.3390/electricity4020009_

Round 1

Reviewer 1 Report

The authors propose a method to forecast the availability of electric car charging stations. The paper is well written, and I only have these few suggestions:

-       Figure 3: I would colour the background of the figure in two different colours to highlight that one part is completed earlier and the other is a repeat whenever a new charging request arrives. Without the figure description, it looks like the model gets re-trained before each request of a charging station comes in.

-       Figure 2, please add what figure ‘a’ and ‘b’ displays in the title and not just in the main text.

-       Table 1, you state that you applied the information to the entire data set. Does this mean that you did not only apply it to the test dataset?

-       You do explain the contribution of the paper very well. To make it easier for the reader to spot the new contributions, it would be good if you highlight a few research questions/objections and answer these in the discussion.

-       I feel that the following sentence is a bit misleading (line 421): “By including charging events at various times and not just from a period when the use of charging stations was impacted by the Corona virus.” I thought for a minute that you had added extra data pre and post pandemic (2019, or late 2022).

Reviewer 2 Report

The authors propose a new model for estimating the charging duration of charging events in real time, which may be used to estimate the waiting time of users at fully occupied charging stations. 

Questions:

1. The article needs a better bibliographic revision. There are many references that are in repositories and have not been published either in conferences or in journals. Also, there are few articles from the last five years.

2. Figures 2a and 2b are not legible and are important. Increase the size of the figures.

3. The differences between the Randon Forest (RF) proposed by the authors and the traditional Random Forest are not clear. It needs to be clearer.

4. The authors inform that they will use the RF as a regression problem but present results of a confusion matrix that originates from a classification problem. This is confusing.

5. What are the minimum and maximum possible values of the R2 index? What are the desired values of the R2 index? Does this index have to be as close to 1?

6. The R2 results presented in Figure 7 indicate that the training stage of the compared traditional methods were not good since R2 assumed negative values. However, the training of the proposed method was good because it assumed positive values. What are the reasons why training in traditional methods was bad?

7. What were the design parameters of the proposed Random Forest and the other techniques that were compared? Did the authors conduct a sensitivity analysis of these parameters and how does this affect the results of the proposed method?

Reviewer 3 Report

I read the article "Data-driven, Short-term Prediction of Charging Station Occupation" with interest and made the following comments:

1) Papers [23] and [24] are not presented and discussed within the article. I therefore propose to the authors to remove them from the list of bibliographic references.

2) In the text associated with Figure 4, the unit of measure for Electric Power must be corrected, kW instead of KW.

3) It would be useful for readers to include the unit of measure for "predictive power" in Tables 2 and 3.

I am convinced that solving these reported issues will not cause problems for the authors and that in this way the quality of the article will improve.

Round 2

Reviewer 2 Report

The authors propose a new model for estimating the charging duration of charging events in real time, which may be used to estimate the waiting time of users at fully occupied charging stations. 

The article has been improved, the contribution is good and all questions have been effectively answered.